# Overcoming Language Priors in Visual Question Answering with Adversarial Regularization

**Sainandan Ramakrishnan**     **Aishwarya Agrawal**     **Stefan Lee**
Georgia Institute of Technology
{sainandancv, aishwarya, steflee}@gatech.edu

## Abstract

Modern Visual Question Answering (VQA) models have been shown to rely heavily on superficial correlations between question and answer words learned during training – *e.g.* overwhelmingly reporting the type of room as kitchen or the sport being played as tennis, irrespective of the image. Most alarmingly, this shortcoming is often not well reflected during evaluation because the same strong priors exist in test distributions; however, a VQA system that fails to ground questions in image content would likely perform poorly in real-world settings.

In this work, we present a novel regularization scheme for VQA that reduces this effect. We introduce a question-only model that takes as input the question encoding from the VQA model and must leverage language biases in order to succeed. We then pose training as an adversarial game between the VQA model and this question-only adversary – discouraging the VQA model from capturing language biases in its question encoding. Further, we leverage this question-only model to estimate the increase in model confidence after considering the image, which we maximize explicitly to encourage visual grounding. Our approach is a model agnostic training procedure and simple to implement. We show empirically that it can improve performance significantly on a bias-sensitive split of the VQA dataset for multiple base models – achieving state-of-the-art on this task. Further, on standard VQA tasks, our approach shows significantly less drop in accuracy compared to existing bias-reducing VQA models.

## 1   Introduction

The task of answering questions about visual content – called Visual Question Answering (VQA) – presents a rich set of artificial intelligence challenges spanning computer vision and natural language processing. Successful VQA models must understand the question posed in natural language, identify relevant entities, object, and relationships in the image, and perform grounded reasoning to deduce the correct answer. In response to these challenges, there has been extensive work on VQA in recent years both in terms of dataset curation [6, 12, 2, 17, 13, 32, 3] and modeling [2, 5, 28, 14, 16, 4, 22, 20].

This widespread interest in VQA has resulted in increasingly sophisticated models achieving higher and higher performance on increasingly large benchmark datasets; however, recent studies have demonstrated that many models tend to have poor image grounding, instead heavily leveraging superficial correlations between questions and answers in the training dataset to answer questions [1, 30, 15, 12]. As a result, these models often exhibit undesirable behaviors – blindly outputting an answer based on first few words in the question (*e.g.* reflexively answering *'What sport ...'* questions with *'tennis'*) or failing to generalize to novel attribute-noun combinations (*e.g.* being unable to identify a *'green hydrant'* despite seeing both hydrants and green objects during training). Perhaps most dissatisfying of all, standard evaluation protocols on benchmark datasets often fail to pick up on these trends due to the presence of the same strong language priors in their test datasets.

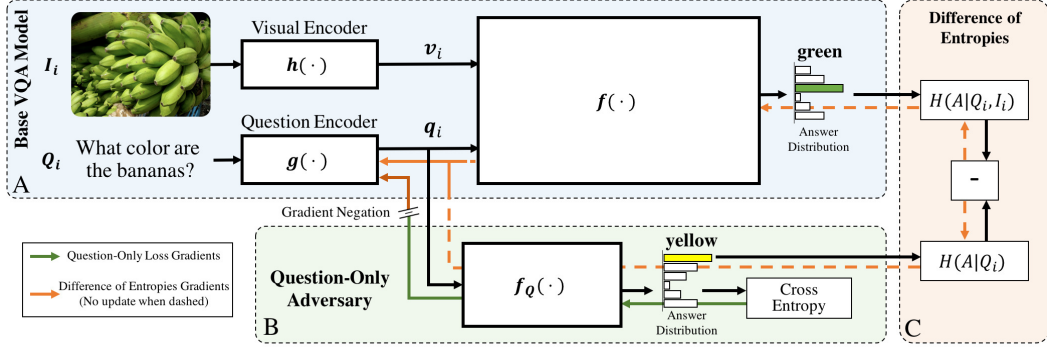

Figure 1: Given an arbitrary base VQA model (**A**), we introduce two regularizers. First, we build a question-only adversary (**B**) that takes the question embedding $\mathbf{q}_i$ from the VQA model and is trained to output the correct answer from this information alone. For this network to succeed, $\mathbf{q}_i$ must capture language biases from the dataset – the same biases that lead the base VQA model to ignore visual content. To reduce these biases, we set the base VQA model and the question-only adversary against each other, with the base VQA network modifying its question embedding to reduce question-only performance (shown here as gradient negation of the question-only model loss) Further, the question-only model allows estimation of the change in answer confidence given image (**C**), which we maximize explicitly.

Recently, Agrawal *et al*. [2] introduced the VQA-CP (Visual Question Answering under Changing Priors) diagnostic split of the VQA [6, 12] dataset to measure the effect of language bias. VQA-CP is constructed such that for each question type (*e.g*. 'What color ...', 'How many ...', *etc*.), the answer distributions vary dramatically between training and test. Consequentially, models with poor grounding and an over-reliance on language priors from the training set fair poorly on this new split. Despite the image still containing all the necessary information to answer the questions, multiple existing VQA models evaluated on this split achieve dramatically deteriorated performance.

One intuitive measure of the strength of language priors in VQA is the performance of a 'blind' model that produces answers given only the question and not the associated image. In fact, this question-only model has become a standard and powerful baseline presented alongside VQA datasets [6, 12, 9, 17]. In this work, we codify this intuition, introducing a novel regularization scheme that sets a base VQA model against a question-only adversary to reduce the impact of language biases.

More concretely, we consider unwanted language bias in VQA to be overly-specific relationships between questions and their likely answers learned from the training dataset – *i.e*. those that could enable a question-only model to achieve relatively high performance without ever seeing an image – and we explicitly optimize the question representation within a base VQA model to be uninformative to a question-only adversary model. In this adversarial regime, the question-only model is trained to answer as accurately as possible given the question encoding provided by the base VQA model; and simultaneously, the base VQA model is trained to adjust its question encoder (often implemented as a recurrent language model) to minimize the performance of the question-only model while maintaining its own VQA accuracy. Moreover, we leverage the question-only model to provide a differentiable notion of image grounding – the change in model confidence after considering the image – which we maximize explicitly for the VQA model. Thus, our objective consists of a question-only adversarial term and a difference of entropies term.

Our approach is largely model agnostic, end-to-end trainable, and simple to implement, consisting of a small, additional classification network built on the question representation of the base VQA model. We experiment on the VQA-CP dataset [2] with multiple base VQA models, and find – 1) our approach provides consistent improvements over all baseline VQA models, 2) our approach outperforms the existing state-of-art grounded-by-design approach [2] significantly, 3) both question-only adversary and the difference of entropies components improve performance and their combination pushes this even further. On standard benchmarks [6, 12] where strong priors from training can be exploited on test set, our approach shows significantly smaller drops in accuracy compared to existing bias-reducing VQA models [2], with some settings facing only insignificant changes.

## 2 Reducing Language Bias Through Adversarial Regularization

Setting aside architectural specifics, the vast majority of VQA models operate on a set of similar design principles – first producing vector representations for the image and question and then combining them to predict the answer (often through complex attention mechanisms). However, when language biases are quite strong, the question feature may already be sufficiently discriminative and the model can learn to ignore the visual signal without facing significant losses during training (*e.g.* "What color is the sky?" always mapping to "blue"). Such a model which fails to ground its answers in the image might be passable for benchmark datasets that carry similar biases; however, in the real-world, where brown grass and gray skies abound, its usefulness would be severely limited. In this section, we address this problem by explicitly reducing the discriminative power of the question feature – introducing a pair of adversarial regularizers that penalize the ability of a separate adversary network to confidently predict the answer from the question encoding alone.

**Preliminaries.** Given a dataset $\mathcal{D} = \{I_i, Q_i, a_i\}_{i=1}^{N}$ consisting of triplets of images $I_i \in \mathcal{I}$, questions $Q_i \in \mathcal{Q}$ and answers $a_i \in \mathcal{A}$, the VQA task is to learn a mapping $F\colon \mathcal{Q} \times \mathcal{I} \to [0,1]^{|\mathcal{A}|}$ which produces an accurate distribution over the answer space given an input question-image pair.

Without loss of generality, we consider differentiable mappings that can be decomposed as an operation $f$ over question and image encodings $g\colon \mathcal{Q} \to \mathbb{R}^d$ and $h\colon \mathcal{I} \to \mathcal{R}^k$ (as shown in Figure 1A). We write the prediction for instance $i$ for this class of models as

$$\mathbf{v}_i = h(I_i), \quad \mathbf{q}_i = g(Q_i)$$
$$P(\mathcal{A} \mid Q_i, I_i) = f\left(\mathbf{v}_i, \mathbf{q}_i\right) \tag{1}$$

where we denote the image and question embeddings as $\mathbf{v}_i$ and $\mathbf{q}_i$ respectively.

Nearly all existing VQA models follow this pattern. The image encoder $h(\cdot)$ is typically a fixed CNN pretrained on either classification or detection and the question encoder $g(\cdot)$ is usually some form of word or character level RNN learned during training. Typically these models are trained with standard cross-entropy, optimizing parameters to minimize (2) over the ground truth data.

$$\mathcal{L}_{VQA}(f, g, h) = \mathbb{E}_{\mathcal{I}, \mathcal{Q}, \mathcal{A}} \left[ -\log P_f(a_i | Q_i, I_i) \right] \approx -\frac{1}{N} \sum_{i=1}^{N} \log f(\mathbf{v_i}, \mathbf{q}_i)[a_i] \tag{2}$$

**Question-Only Model.** One intuitive measure of the power of language priors in VQA is the ability of a model to make low-error answer predictions from the question alone – in fact, some form of this 'blind' model has been frequently presented alongside VQA datasets for exactly this purpose [6, 12, 9, 17]. We formalize this question-only model as a mapping $f_Q$. As above, we assume $f_Q$ is differentiable and operates on learned question encodings such that $f_Q$ makes predictions

$$P_{f_Q}(\mathcal{A} \mid Q_i) = f_Q(\mathbf{q}_i), \quad \mathbf{q}_i = g(Q_i). \tag{3}$$

We parameterize this model as a simple two-layer neural network but note that arbitrary choices can be made in this regard. As above, this model can be trained with cross-entropy, minimizing

$$\mathcal{L}_{QA}(f_Q, g) = \mathbb{E}_{\mathcal{Q}, \mathcal{A}} \left[ -\log P_{f_Q}(a_i | Q_i) \right] \approx -\frac{1}{N} \sum_{i=1}^{N} \log f_Q(\mathbf{q}_i)[a_i]. \tag{4}$$

### 2.1 Adversarial Regularization with a Question-Only Adversary

For any model of the form presented in (1), we can now introduce a simple adversarial regularizer that explicitly reduces the effect of language biases by modifying the question encoder to minimize the performance of this question-only adversary. Specifically, given a VQA model decomposed as $f, g, h$, we splice on the question-only model $f_Q$ such that $f_Q$ takes as input the encodings produced by $g(\cdot)$ (as in Figure 1), and establish opposing losses for the two networks which we detail below.

**Learning the Question-Only Adversary.** The question-only model $f_Q$ is trained to minimize the cross-entropy loss $\mathcal{L}_Q$ in (4); however, parameters in $g(\cdot)$ are not updated with respect to this loss – in effect, this forces $f_Q$ to perform as well as possible given the question encodings produced by question encoder $g(\cdot)$ from the base VQA model.

**Adversarial Regularization for VQA.** As performance of the question-only model acts as a proxy for the language biases represented in the question encodings $\mathbf{q}_i = g(Q_i)$, one approach to reduce bias representation is to adjust $g(\cdot)$ such that the question-only model does poorly. As such, we can write this adversarial relationship between the question-only ($f_Q$) and base VQA models ($f, g, h$) as

$$\min_{f,g,h} \max_{f_Q} \mathcal{L}_{VQA}(f,g,h) - \lambda_Q \mathcal{L}_{QA}(f_Q, g) \tag{5}$$

We note that in practice, training with this adversarial regularizer can be realized with a simple gradient negation of the question-only adversary's loss as shown in Figure 1. Specifically, we back-propagate the negative of the gradient of $\mathcal{L}_Q(f_Q, g)$ accumulated at $\mathbf{q}_i$ through the question encoder – updating the question encoder in a way that maximizes $\mathcal{L}_Q(f_Q, g)$.

The regularization coefficient $\lambda_Q \geq 0$ in (5) controls the trade-off between VQA performance and language bias reduction. For low values of $\lambda_Q$, little regularization occurs and the base model continues to learn language priors. On the other hand, large values of $\lambda_Q$ force the model to remove all discriminative language biases, resulting in poor VQA performance for both the base VQA model and the question-only adversary – essentially stripping the question encoding of even basic question-type information (*e.g.* failing to learn that *"What color ... ?"* questions require color answers).

## 2.2 An Adversarial Difference of Entropies Regularizer

As the effect of this over-regularization for high-values of $\lambda_Q$ highlights, the question-only adversary does not capture the full nuance of language bias in VQA. Given the question *"What color is the sky?"* it is reasonable to have a prior that the answer may be *"blue"*, but critically this belief should update depending on observations – *i.e.* the answer distribution should sharpen after viewing the image.

To capture this intuition, we add an additional term that aims to maximize the information gained about the answer from looking at the image. Specifically, we introduce another adversarial regularizer corresponding to the difference in entropies between the base model prediction given the image and the question-only model which we write as

$$
\begin{aligned}
\mathcal{L}_H(f, g, h, f_Q) &= \mathbb{E}_{I,Q}\left[H(\mathcal{A} \mid \mathcal{Q}) - H(\mathcal{A} \mid \mathcal{I}, \mathcal{Q})\right] \tag{6}\\
&= \mathbb{E}_{q\sim P(\mathcal{Q})}\left[H(\mathcal{A} \mid q)\right] - \mathbb{E}_{q,v\sim P(\mathcal{Q},\mathcal{I})}\left[H(\mathcal{A} \mid q, v)\right] \tag{7}\\
&\approx \frac{1}{N}\sum_{i=1}^{N}\left( H\left(f_Q(\mathbf{q}_i)\right) - H\left(f(\mathbf{v_i}, \mathbf{q}_i)\right) \right) \tag{8}
\end{aligned}
$$

We note that this regularizer resembles the conditional mutual information (CMI) between the answer and image given the question $I(A; I|Q)$; however, $f_Q(q)$ is not constrained to be the marginal of $f(v, q)$ such that estimating the CMI in this way is ill-defined.

We can then update the adversarial relationship between $f$ and $f_Q$ from (5) with $\mathcal{L}_{MI}$, writing

$$\min_{f,g,h} \max_{f_Q} L_{VQA}(f,g,h) - \lambda_Q \mathcal{L}_{QA}(f_Q, g) - \lambda_H \mathcal{L}_H(f,g,h,f_Q) \tag{9}$$

where $\lambda_H \geq 0$ controls the strength of the difference of entropies regularizer. Note that while $\mathcal{L}_H$ is a function of $f$, $g$, $h$, and $f_Q$, we only update the parameters of the question encoding $g$ based on this loss. Otherwise, $f_Q$ could learn to produce sharp output distributions from arbitrary question features to minimize $\mathcal{L}_H$. Likewise, $f$ or $h$ can easily adjust to produce arbitrarily peaky outputs, which we observe can lead to significant over-fitting.

As before, the question-only adversary $f_Q$ in this setting must still perform as well as possible given the question embedding from $g(\cdot)$, but this embedding is now additionally adjusted to maximize the entropy of $f_Q$'s output, while minimizing that of the VQA model. In the experiments that follow, we show that both of these adversarial regularizers improve performance on a language bias sensitive task. Further, we note that their benefits compound, with models combining both terms performing better across a wider range of regularization coefficients.

## 3 Related Work

Essentially all real world datasets have some form of bias either due to their collection process (*e.g.* reporting biases [11]) or those reflecting real-world human biases (*e.g.* capturing stereotypical gender

roles). These biases are often less than subtle, with human annotators easily identifying from which dataset specific instances originate [26] on sight. In this section, we discuss related work on bias in VQA, how to reduce it, and on adversarial training regimes related to our approach.

**Language Bias in VQA.** In VQA, a significant source of bias is the strong association between question words and answers (*e.g.* 'Is there a ...' questions predominantly being answered with 'Yes' in VQA v1 [6]). Building off [6], Goyal *et al.* [12] introduced the VQA v2 dataset which significantly weakened language priors in the VQA v1 dataset. For each VQA v1 question, VQA v2 was constructed to also contain an image which is similar to the VQA v1 image, but has a different answer to the same question – effectively reducing the sharpness of question-only priors. However, even with this additional 'balancing' there exist significant biases in the dataset. In these works, the extent of language biases was measured through a baseline which must predict answers from questions alone. Deriving inspiration from this baseline, we introduce a question-only adversary to explicitly reduce the ability of the question-only baseline to predict answers from questions alone.

Recently, Agrawal *et al.* [2] introduced the VQA-CP (VQA under Changing Priors) dataset, a diagnostic split of the VQA datasets [6, 12] that is constructed with vastly different answer distributions between train and test. Consequentially, models that overly rely on language biases or have poor visual grounding do poorly on this split, with [2] reporting dramatic drops in performance for state-of-the-art VQA models. We use VQA-CP as a testbed for our adversarial regularization approach and show consistent improvements over base models and existing work.

**Overcoming Unwanted Biases.** In addition to proposing the VQA-CP dataset, [2] designed a Grounded VQA model (GVQA) that includes hand-designed architectural restrictions to prevent the model from exploiting language correlations in training data. Specifically, GVQA disentangles the visual concepts in the image that need to be recognized, from the space of plausible answers – introducing separate visual concept and answer cluster classifiers. While the model performs well on VQA-CP, it is complicated in design and requires training multiple stages in sequence. In contrast, our approach is implemented as a simple drop-in regularizer built on top of existing VQA models and enables end-to-end training without changing the underlying model architecture, unlike the design principles of GVQA which require significant architecture adjustments if extended to new models. Further, we find our approach significantly outperforms the hand-crafted network structure of GVQA.

Similarly, neural module network style architectures [5, 14, 16] introduce an explicit structure in the model that separates the question from the reasoning on the image. These models predict the layout of modular computational units from the question content and these modules then operate on the image to produce an answer. Despite this explicitly compositional reasoning process, these models also suffer a dramatic drop in performance when evaluated on VQA-CP [2]. In contrast, our proposed approach performs well on VQA-CP and can be applied to any model architecture.

In recent work, Burns *et al.* [7] investigate the generation of gender-specific words in image descriptions which is often skewed in captioning models (*e.g.* models nearly always using male words to describe snowboarders). The proposed approach encourages the model to confidently predict gendered words when gender information is visually present and to be unsure when it is occluded by a mask. While effective, this model requires segmentation of the visual concept of interest.

More generally, Zhao *et al.* [31] address the issue of bias amplification broadly, introducing a inference-time procedure to recalibrate the model. However, this approach requires computing the output distribution for each element of a test set before this procedure can be performed. In comparison, we present a training-time procedure that results in a less biased model. In principle, [31] could also be applied to a model trained under our proposed regularizers.

**Adversarial Learning.** Generative Adversarial Networks (GANs) [10] have received significant recent interest for their ability to model complex distributions – finding use in a variety of image and language generation tasks [10, 23, 29, 8, 21]. Recently, other adversarial training schemes have been proposed to encourage various forms of invariance in intermediate model representations [18, 19, 27].

Most related to our approach, Lample *et al.* [18] introduce an autoencoder framework with an adversarial loss for attribute-based image manipulation. Given an input image and a set of attributes (*e.g.* a photo of a person and their gender or age), the task is to manipulate the image such that it has the desired attributes. Unfortunately, without multiple pairings of the same image with different attributes, it is challenging to learn disentangled image representations that generalize to new input-attribute combinations. An adversarial model is introduced that is trained to predict attributes from the

input image encoding alone. In combating this adversary, the image encoder model learns to produce attribute invariant image encodings. This improves generalization by forcing the attribute-augmented decoder to meaningfully rely on input attributes to accurately reproduce input images.

Similarly, our question-only adversary encourages the VQA question encoder to remove answer-discriminative features from the question representation. However, breaking the parallels with [18], the answer themselves are not added back as inputs to controllably recondition the model on these features. Rather, the VQA model must rely on the combination of question and image features to recover the answer information. In this way, the language-level answer information (*e.g.* that most grass is green) is removed from the question and instance-specific information from the image must be used instead. We take this notion further by leveraging the question-only adversary to estimate and directly maximize the change in confidence after observing the image, which we show provides substantial benefits when paired with the question-only adversary.

## 4 Experiments

**Implementation.** Our question-only adversary model is implemented as a 2-layer multi-layer perceptron with 256 hidden units and a ReLU activation that takes as input the question encoding from a base VQA network. The network's output is a distribution over the candidate answers. We train the entire system (base VQA and question-only model) end-to-end with parameters initialized from scratch. We set batch size to 150, learning rate to 0.001, weight decay of 0.999 and use the Adam optimizer. The model takes ∼8 hours to train on a TITAN X for SAN (Torch, ∼60 epochs) and < 1 hour for UpDown (PyTorch, ∼40 epochs). We use public codebases for both.

As discussed in Section 2, we update the parameters of the question encoding with respect to the VQA loss, the difference of entropies loss, and the negative of the question-only loss. The remaining VQA model parameters are trained with just the VQA loss. The question-only model is updated only by its VQA loss cross entropy loss term despite contributing to the difference of entropies loss.

**Models.** We evaluate the effect of our proposed regularization on the following base models:

- **Stacked Attention Network (SAN) [28]** – SAN encodes questions with a long short-term memory (LSTM) encoder and the image is encoded with a pretrained VGGNet [25]. The model performs two-hop question-based image attention and the final joint feature is passed to a 1000-way answer classifier. This model is trained with standard cross-entropy.

- **Bottom-Up and Top-Down Attention (UpDn) [4]** – Up-Down encodes questions with a gated recurrent unit (GRU) encoder and represents images as a set of bounding box features extracted from Faster R-CNN [24]. Soft-attention over these regions is computed based on the question features and the attention-pooled feature is combined with the question as input to the classification layer. This model is trained directly on VQA score under a multi-label binary cross-entropy loss (see [4] for more details). We also apply this loss for the question-only model in our experiments, but compute a softmax over these outputs when computing entropies.

For both SAN[1] and Up-Down[2], we build on top of publicly available reimplementations. In the following results, we indicate the addition of our question-only adversarial regularization with **Q-Adv** and the difference of entropies term as **DoE**.

We also compare to the **GVQA** [2] model built atop **SAN** and introduced alongside the VQA-CP dataset. GVQA explicitly separates perception from question answering by introducing a Visual Concept Classifier (VCC) and an Answer Cluster Predictor (ACP). The VCC is a bank of pretrained classifiers for visual entities and attributes and its output is modulated by the ACP. The ACP takes a question and predicts one of a predefined answer clusters. The ACP masked VCC outputs are used to predict the answer. A separate branch handles binary questions as a visual verification task. By design, this model isolates the answering module from the input question, mitigating the effect of language biases, but at a cost of relatively low standard VQA performance and multi-stage training.

**Datasets and Evaluation.** We train our models on the VQA-CP [2] train split and evaluate on the test set using the standard VQA evaluation metric [6]. For each model, we also report results when trained

Table 1: Performance on VQA-CP v2 `test` and VQA v2 `val`. We significantly improve the accuracy of base models and achieve state-of-the-art performance on the VQA-CP dataset.

| | Model | $\lambda_Q$ | $\lambda_H$ | VQA-CP v2 test | | | | VQA v2 val | | | |
| --- | --- | --- | --- | --- | --- | --- | --- | --- | --- | --- | --- |
| | | | | Overall | Yes/No | Number | Other | Overall | Yes/No | Number | Other |
| | GVQA [2] | - | - | 31.30 | 57.99 | 13.68 | 22.14 | 48.24 | 72.03 | 31.17 | 34.65 |
| | SAN [28] | - | - | 24.96 | 38.35 | 11.14 | 21.74 | 52.41 | 70.06 | 39.28 | 47.84 |
| Ours | SAN + Q-Adv | 0.15 | - | 27.24 | 54.50 | 14.91 | 16.33 | 52.18 | 69.81 | 39.21 | 47.52 |
| | SAN + DoE | - | 25 | 25.75 | 42.21 | 12.08 | 20.87 | 52.38 | 70.05 | 39.64 | 47.41 |
| | SAN + Q-Adv + DoE | 0.15 | 25 | **33.29** | **56.65** | **15.22** | **26.02** | 52.31 | 69.98 | 39.33 | 47.63 |
| | UpDn [4] | - | - | 39.74 | 42.27 | 11.93 | 46.05 | 63.48 | 81.18 | 42.14 | 55.66 |
| Ours | UpDn + Q-Adv | 0.005 | - | 40.08 | 42.34 | 13.02 | 46.33 | 60.53 | 77.70 | 41.00 | 52.65 |
| | UpDn + DoE | - | 0.05 | 40.43 | 42.62 | 12.19 | **47.03** | 63.43 | 81.15 | 42.64 | 55.45 |
| | UpDn + Q-Adv + DoE | 0.005 | 0.05 | **41.17** | **65.49** | **15.48** | 35.48 | 62.75 | 79.84 | 42.35 | 55.16 |

and evaluated on the standard VQA train and validation splits [6, 12] with the same regularization coefficients used for VQA-CP to compare with [2].

VQA-CP does not have a validation set and generating such a split is complicated by the need for it to contain priors different from both the training and test sets in order to be an accurate estimate of generalization under changing priors – an ill-defined notion for binary questions. As such, we set initial regularizer coefficients such that gradients at the question encoding are roughly equal in magnitude for all loss terms at the beginning of training and then explore a small region around this point. We report the best performing coefficients alongside our results and provide further analysis of the effect of these parameters in Section 5. Notably, we find these coefficients to be highly model dependent but generalize well between datasets and regularizer ablations. All models are trained until convergence as we have no validation set on which to base early-stopping.

## 5 Results

Table 1 presents our primary results on both the VQA-CP v2 and the VQA v2 datasets. Table 2 also shows limited results on the much more biased VQA v1 dataset [6] and its CP counterpart – VQA-CP v1 [2]. We make a number of observations below.

**The proposed regularizers help, resulting in state-of-art performance on VQA-CP.** For both SAN and UpDn models, adding the question-only adversary (Q-Adv) improves the performance of the respective base models (2.28% for SAN and 0.34% for UpDn) on the VQA-CP v2 dataset. Similarly, the difference of entropies (DoE) regularizer boosts the performance of both SAN and UpDn models, gaining improvements of 0.79% and 0.69% respectively. The combination of the Q-Adv and DoE regularizers further boosts the performance, resulting in 8.33% improvement over SAN and 1.43% over UpDn. Comparing our SAN + Q-Adv + DoE model to GVQA which is also built on top of SAN, we outperform GVQA significantly (1.99%). Our UpDn + Q-Only + DoE model also sets a new state-of-the-art on VQA-CP v2, improving over GVQA by 9.87% (although it is important to note the more powerful base architecture already outperforms GVQA by 8.44%).

Similar trends repeat for VQA-CP v1 as well. With the question-only regularizer improving SAN by 1.14%, DoE by 0.95%, and their combination by over 16.55% – outperforming GVQA by 4.2% and again setting state-of-the-art. We note that these larger gains are in part due to the increased language biases present in the VQA-CP v1 dataset.

Moreover, we find the question-only network performs increasingly poorly as our models perform better on VQA-CP – indicating that optimization is going well and that the intuition behind our regularizers seems well-founded. For quantitative results, see the supplementary.

**The proposed regularizers do not hurt significantly on VQA v2.** When trained and tested on the VQA v2 dataset (right side of Table 1), the addition of the proposed regularizers results in a insignificant drop in the performance for SAN (0.1%) and a minor drop in performance for UpDn (0.73%) compared to prior work. This is in contrast to GVQA, whose performance drops by 4.17% for SAN on VQA v2 (note that GVQA is built off of SAN). For completeness we further evaluate on

Table 2: Performance on VQA-CP v1 test and VQA v1 val.

| Model | $\lambda_Q$ | $\lambda_H$ | VQA-CP v1 test | | | | VQA v1 val | | | |
|---|---|---|---|---|---|---|---|---|---|---|
| | | | Overall | Yes/No | Number | Other | Overall | Yes/No | Number | Other |
| GVQA [2] | - | - | 39.23 | 64.72 | 11.87 | 24.86 | 51.12 | 76.90 | 32.79 | 36.43 |
| SAN [28] | - | - | 26.88 | 35.34 | 11.34 | 24.70 | 55.86 | 78.54 | 33.46 | 44.51 |
| SAN + Q-Adv | 0.15 | - | 28.02 | 35.70 | 11.70 | 19.99 | 52.01 | 70.68 | 32.39 | 42.91 |
| SAN + DoE | - | 25 | 27.83 | 36.33 | 11.15 | 24.03 | 54.08 | 78.19 | 32.59 | 41.44 |
| SAN + Q-Adv + DoE | 0.15 | 25 | **43.43** | **74.16** | **12.44** | **25.32** | 52.15 | 71.06 | 32.59 | 42.91 |

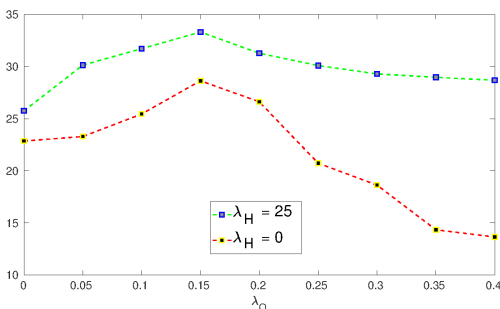

Figure 2: Maximizing difference of entropies (DoE) along with the question-only adversarial regularization for the SAN model, not only improves results on changing priors, but also stabilizes training.

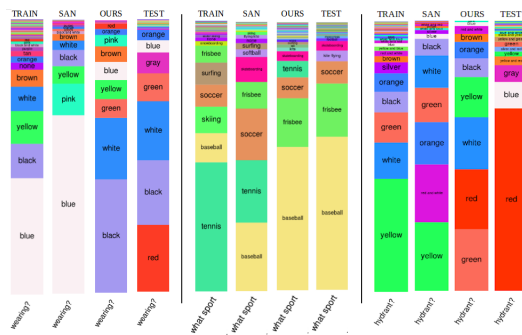

Figure 3: Answer distribution for SAN+Q-Adv+DoE mimic the prior less for questions with high language bias.

VQA v2 test-std, finding our SAN+Q-Adv+DoE model gives 52.95% overall accuracy, 2.33% less than base SAN. Results on this split were not reported for GVQA in [2].

**The more the biases, the higher the gain on VQA-CP, and the higher the loss on VQA.** VQA v1 has significantly more bias than VQA v2 and consequentially VQA-CP v1 has a sharper change between training and test. As such, we observe the proposed regularizers improve over the base model significantly more in VQA-CP v1 (16.55% for SAN) than in VQA-CP v2 (8.33% for SAN). For the same reasons, the proposed regularizers hurt a bit more on VQA v1 (3.71% for SAN compared to 0.1% on VQA v2), where strong language biases can be leveraged to boost performance. However, this drop in the performance on VQA v1 is still significantly less than that with GVQA (4.74%). We also found that the proposed approach has strengths complementary to SAN (see supplementary).

**UpDn [4] is less driven by biases than SAN.** The drop in the performance of UpDn from VQA v2 to VQA-CP v2 is 23.74% which is significantly less than that of SAN (27.45%). This shows that UpDn may be less driven by biases than SAN. And hence, the gains in UpDn (1.43%) due to the proposed regularizers are less than those in SAN (8.33%).

**Our approach results in less biased output distributions.** Figure 3 shows answer frequency distributions for VQA v2 train, SAN, our SAN+Q-Adv+DoE model (marked Ours), and VQA v2 test for three questions: *"What color is the dress she/he is wearing?"*, *"What sport ...?" "What color is the fire hydrant?"*. It is quite clear that while neither of the SAN based models completely match the test distribution, the base SAN model aligns significantly more with the training distribution – even amplifying the bias for 'blue' in the first question despite very few answers being 'blue' in test.

**Difference of entropies (DoE) stabilizes training with the question-only adversary.** Figure 2 shows VQA-CP v2 test performance of the SAN model, for a range of question-only regularizer coefficients $\lambda_Q$. We can see that when the DoE term is not used (orange line), performance begins to drop after approximately 0.2 and by 0.35 has deteriorated significantly. At these higher values, nearly all discriminative information in the question encoding is lost – with the VQA model sacrificing its own performance to lower that of the question-only model. However, we observe that for reasonable values of $\lambda_H$, the strength of the question-only adversary can be varied over a much wider range with less dramatic losses (blue curve in Figure 2). We observe a similar trend when keeping $\lambda_Q$ constant and sweeping over $\lambda_H$, wherein a dramatic improvement is observed when moving to non-zero $\lambda_H$

and then a slow decay for large values of $\lambda_H$. Unlike the question-only adversary, the DoE regularizer simultaneously seeks to sharpen the VQA models posterior while weakening the question-only prior.

**Question-only performance:** We study the performance of the question-only model after being trained on VQA-CP v2 using our regularizers. We compare to a question-only model trained without these regularizers, i.e. a model trained to predict the correct answer given the question-encoding learned by the base VQA model. We find this Q-only(SAN) model achieves 24.84% on the VQA-CP v2 training set compared to 13.85% for our SAN+Q-only+DoE model, demonstrating that our approach has effectively restricted the discriminative information in the question encoding.

**Proposed model shows complementary strengths with the base model:** To study whether our models learn complementary strengths to the base VQA models, we experiment with ensembles of both models. First, we consider oracle ensembles where the best model output for each data point is considered for evaluation. This is an upper bound on ensemble performance that relies on knowing ground truth. We find that the Oracle(Ours, SAN) ensemble outperforms two separately trained SAN models Oracle(SAN, SAN), by 1.48% for VQA v1 and by 3.46% for VQA v2– significantly lower gains than with Oracle(GVQA, SAN) which improves by 5.28%. It is notable however that the architecture of GVQA is significantly different from the base SAN model and hence is expected to exhibit different error patterns and a higher Oracle accuracy. To take a more attainable view, we also computed a standard ensemble Ensemble(Ours, SAN) and compared to an Ensemble(SAN, SAN) model, outperforming it by 1.24% for VQA v2 but falling short by 0.15% for VQA v1. In contrast, Ensemble(GVQA, SAN) improves VQA v2 performance by only 0.54%.

**Qualitative Examples:** Figure 4 shows example outputs and heatmaps for the SAN model with and without our regularizers on VQA-CP v2. In addition to improving accuracy, our regularized approach often results in repositioned heatmaps (surfer bottom right).

# 6   Conclusion

We propose a novel adversarial regularization scheme for reducing the memorization of dataset biases in VQA based on a question-only adversary and the difference of model confidences after processing the image. Experiments on the VQA-CP dataset, show that this technique allows existing VQA models to significantly improve performance in the midst of changing priors. Consequently, we achieve state-of-the-art performance on VQA-CP. Our approach can be implemented as a simple, drop-in module on top of existing VQA models and easily trained end-to-end from scratch.

**Acknowledgements** This work was supported in part by NSF, AFRL, DARPA, Siemens, Google, Amazon, ONR YIPs and ONR Grants N00014-16-1-{2713,2793}. The views and conclusions contained herein are those of the authors and should not be interpreted as necessarily representing the official policies or endorsements, either expressed or implied, of any sponsor.

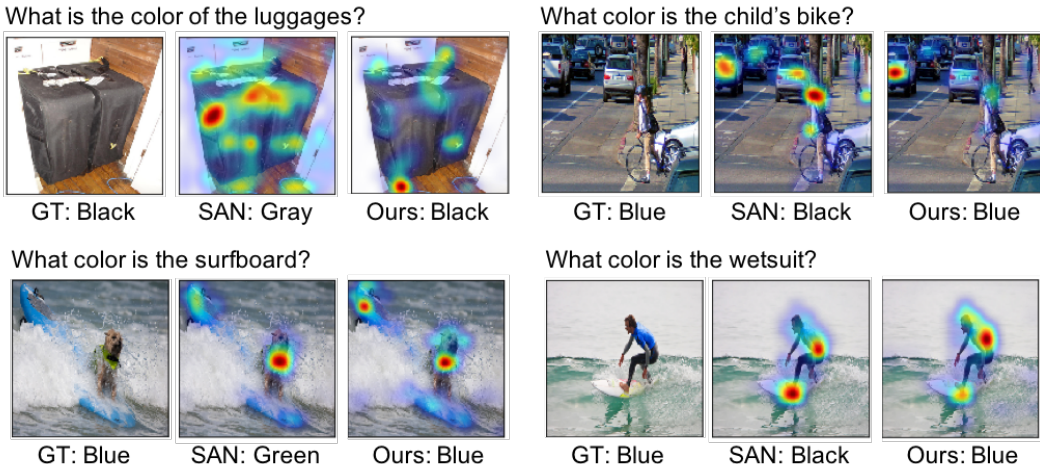

Figure 4: Qualitative examples of outputs and attention maps for SAN with (Our) and without (SAN) our proposed regularizers on VQA-CP v2.

## Footnotes

[1]SAN Codebase: https://github.com/abhshkdz/neural-vqa-attention

[2]Up-Down Codebase: https://github.com/hengyuan-hu/bottom-up-attention-vqa

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
