[Reviews · NeurIPS 2018]

Reviewer 1



This paper studies the problem of handling the langauge/text pariors in the task visual question answering. The great performance achieved by many state-of-the-art VQA systems are accomplished by heavily learning a better question encoding to better capture the correlations between the questions and answers, but ignore the image information. So the problem is important to the VQA research community. In general, the paper is well-written and easy to follow. And some concerns and sugggestions can be found as the following: 1) The major concern is the basic intuition of the question-only adversary: The question encoding q_i from the question encoder is not necessarily the same bias that lead the VQA model f to ignore the visual content. Since f can be a deep neutral network, for example, deep RNN or deep RNN-CNN to leverage both the question embedding and visual embedding, thus the non-linearity in f would make the question embedding as a image-aware represention to generate the answer distribution. Thus if the f_Q and f is different as stated in the paper (since the VQA models f are SAN and UpDn, the f_Q is a 2-layer dense network), then the question embedding equals to q_i (since it is linear transformation throught through the 2-layer dense network) to generate the answer distribution for the question-only adversary. Then the question embedding for question-only mdoel and the VQA model are different, which make the question-only model in the adversarial scheme doesn't hold. 2) The mutual information reguarlizer seems needs to be re-designed. The current optimization direction will let question encoding contains more information about the image. However this intuition doesn't mean the question encoding contain letter language priors. Adding anther mutual information regularizier to minimize the H(A|q_i) could help. 3) In the experiment, the question-only model can be better by introducing non-linearity to make the quesiton encoding behave similarity compared to the VQA models. 4) How to train the question-only model? It is trained based on equation (4) or (9)? 5) How to tune the lambda_q and lambda_mi? As the numbers in Table1 seems quite effective, but not very intuitive. 6) In table 1, why the gain of the VQA+MI performs the best on the VQA v2 dataset amony the versions with regularizers? Since by adding the mutual information regualrizer, the answers could better leverage the image information, it could generate slightly better result compared to the base models.

Reviewer 2



Bias in Visual Question Answering dataset can harm the image-grounding of the model, by relaying on the question modality, and prone to failure in the general real-life case. The paper deals with this issue by discouraging the VQA model from capturing language biases in its question encoding, alternating the question embedding to reduce question-only model in an adversarial setup. Moreover, the paper deals with the nuance of language bias in VQA, by encouraging visual grounding via mutual information of image and answer given the question. Strengths: • Compared to GVQA method, the adversarial loss is simpler and more elegant. • The mutual information between answer and image is interesting. To my knowledge this is original component. It captures the nuance in question bias for VQA, whether the bias is attributed to annotations, or it is a real-world bias, e.g, sky is blue. This component improves the results and also stabilize training over different hyper-parameters. • Comprehensive evaluation on SOTA UpDn, and not only SAN. Also, evaluation on both VQA v2 and VQA v1. • The paper is well organized and easy to read, the contributions clearly laid out. Weaknesses: • Evaluation on Oracle(Ours, SAN) reported only in the supplementary. In my opinion, it should be part of the paper. To me, this is the most significant score, significant model should be useful to improve the test score of VQA2.0 dataset, which is still far from being solved. If I understand correctly from the supplementary, your model improves by 3.46%, while Oracle(GVQA, SAN) by 4.42%. The paper should discuss weaknesses as well. • A diagnose of stronger f_Q networks will be interesting. A 2-NN network is way too naïve. Why not trying some sort of LSTM variation, which is the common practice? • I would appreciate qualitative analysis, i.e, showing images that your model succeeds on and GVQA failed. • Technical information is missing, size of train/test split, training parameters: optimization method used, time of training, hardware, number of parameters for each model, framework used. • I wonder, is it possible to run your model on the train+val/test split of VQA2.0, which usually not correlates with score on train/val split. Conclusion: The paper discuss an interesting loss to deal with vqa bias while training. While it works better on VQA-CP v2, surprisingly the Oracle version is weaker than GVQA, a better discussion is expected. Also technical information given is limited. After author response: I'm happy the authors agreed to add suggested important results to the revised version. I believe this paper will be a good contribution to the conference, and hopefully will enlighten a future research related to bias in QA tasks.

Reviewer 3



In this paper the authors propose an interesting idea where they aim to overcome overfitting of VQA models to the dataset by using adversarial regularization. The regularization is obtained through two different ways, one by regularizing adversarially on a question only model and second by regularizing by maximizing the answer model to improve by observing the image. The method proposed seems to be correct and the formulation makes sense. The evaluation shows improvements over SAN (Das et al) and Up-Down models (Anderson et al) for the VQA-CP dataset. The only problem is that the results provided in supplementary are not clear. From what I could understand, the question only models and the ones improved by regularization still perform inferior to question only model even in the VQA-CP v2 dataset. This implies that the method fails on VQA-CP v2 dataset? I am probably missing something here. It would be very useful if the authors could clarify the results provided in supplementary during rebuttal. Further, would the method improve by incorporating a separate attention based regularization as well? -- After author feedback The feedback addresses the points raised in the review satisfactorily.